# Cohort profile: the United Kingdom Childhood Cancer Study (UKCCS) – a UK-wide population-based study examining the health of cancer survivors

Eve Roman ![ORCID],[1] Eleanor Kane ![ORCID],[1] Alexandra Smith ![ORCID],[1] Debra Howell ![ORCID],[1] Rebecca Sheridan ![ORCID],[1] Jill Simpson,[1] Audrey Bonaventure,[2] Sally Kinsey[3]

¹Department of Health Sciences, University of York, York, UK
²INSERM, Université Paris Cité and Université Sorbonne Paris Nord, Paris, Île-de-France, France
³Paediatric Haematology, Leeds Children's Hospital, Leeds, UK

**Correspondence to**
Professor Eve Roman;
eve.roman@york.ac.uk

## ABSTRACT

**Purpose** The United Kingdom Childhood Cancer Study's (UKCCS's) matched cohort was established to examine the longer term morbidity and mortality of individuals previously diagnosed with cancer before 15 years of age, comparing future healthcare patterns in 5-year cancer survivors to baseline activity seen in age- and sex-matched individuals from the general population.

**Participants** Predicated on a national childhood cancer case-control study conducted in the early 1990s (4430 cases, 9753 controls) in England, Scotland and Wales, the case population comprises 3125 cancer survivors (>5 years), and the control population 7156 age- and sex-matched individuals from the general population who did not have cancer as a child. Participants are now being followed up via linkage to national administrative healthcare databases (deaths, cancers and secondary care hospital activity).

**Findings to date** Enabling the creation of cohorts with minimal selection bias and loss to follow-up, the original case-control study registered all newly diagnosed cases of childhood cancer and their corresponding controls, regardless of their family's participation. Early findings based on the registered case population found marked survival variations with age and sex across subtypes and differences with deprivation among acute lymphoblastic leukaemia (ALL) survivors. More recently, comparing the health-activity patterns of the case and control populations revealed that survivors of childhood ALL experienced excess outpatient and inpatient activity across their teenage/young adult years. Adding to increased risks of cancer and death and involving most clinical specialties, excesses were not related to routine follow-up monitoring and showed no signs of diminishing over time.

**Future plans** With annual linkage updates, the UKCCS's maturing population-based matched cohorts provide the foundation for tracking the health of individuals through their lifetime. Comparing the experience of childhood cancer survivors to that of unaffected general-population counterparts, this will include examining subsequent morbidity and mortality, secondary care hospital activity and the impact of deprivation on longer term outcomes.

## STRENGTHS AND LIMITATIONS OF THIS STUDY

⇒ Covering England, Scotland and Wales, findings from this national study are based on routinely compiled linked health data, not self-report.

⇒ Regardless of whether or not their families participated in the original case-control study, all children are included in the matched cohort, enabling the investigation of selection bias.

⇒ With minimal selection bias and loss to follow-up, the individually age- and sex-matched control population has a similar deprivation distribution to that of the case population, providing a robust baseline against which to evaluate the impact of deprivation across the life-course.

⇒ Analyses are constrained by the fact that national administrative healthcare data are primarily collected for administrative and clinical purposes and not for research.

⇒ With respect to weaknesses, lack of primary care data and information from psychosocial services are obvious deficiencies that currently affect all UK record-linkage studies of the type described here.

## INTRODUCTION

In recent decades, advances in molecular biology and therapy have transformed the landscape for several childhood cancers, changing many from rapidly fatal diseases to treatable conditions with a good prognosis; the overall survival now exceeds 80% in the UK and other economically developed countries.[1 2] Accordingly, it is estimated that by 2030, there will be around three-quarters of a million childhood cancer survivors in Europe alone; this number increasing as therapies improve and populations age.[3] While the majority of cancers in children are more responsive to chemotherapy than those in adults, treatments are often more aggressive, and adverse health problems are well known to

**BMJ**

occur later in life (eg, second cancers, cardiac and bone problems and fertility issues).[4–7]

With a view to investigating childhood cancer survivors' health, the national matched cohort described in this report was established with the aim of examining the relationship between cancers diagnosed before 15 years of age and other morbidities and health states in the years after cancer diagnosis; specific aims include the examination of secondary care hospital activity patterns and the impact of deprivation on longer term outcomes. The cohorts are predicated on the structures established in the United Kingdom Childhood Cancer Study (UKCCS), which was set up as a national population-based case-control study in the early 1990s in England, Scotland and Wales to investigate a wide range of possible causes of childhood cancer. Collecting data from multiple sources (including interviews with parents, primary care records, obstetric/neonatal notes, birth certificates, household radiation measurements and pretreatment and remission blood samples), the case-control study investigated the potentially carcinogenic effects of a wide range of physical (eg, non-ionising radiation), chemical (eg, drugs) and biological (eg, infectious agents) agents.[8–14] Examining associations in the prenatal, in utero and postnatal periods, as well as associations with birth characteristics (eg, birth weight) and other illnesses (eg, allergies), over 90 reports have thus far been published (www.UKCCS.org).

Underpinned by updated ethics and legal permissions, the UKCCS has now been transformed into a matched population-based cohort, with follow-up linkages to electronic national administrative healthcare databases (inpatient and outpatient hospital episode statistics—HES, cancers and deaths).[15–17] Importantly, all cases and controls (regardless of parents' participation in the original case-control study) are being tracked forwards in time via annually updated linkages; the comparison cohort (controls) enabling robust baseline effect measures to be estimated. Currently, over 25 years of follow-up data are available with which to investigate healthcare patterns and health events occurring in cancer survivors. The children in the original UKCCS are presently aged between 27 and 46 years, and this report describes the study's underpinning cohort methods and summarises some of the initial findings.

## COHORT DESCRIPTION

Full details of the UKCCS case-control study (www.UKCCS.org), which provides the foundation of the matched cohort described here, have been published.[8 10 18] Briefly, the sampling frame for cases and controls comprised all children registered with the National Health Service (NHS) in England, Scotland or Wales at the time the study was conducted (>98% of the childhood population in these countries); unfortunately, for logistical reasons Northern Ireland (~3.5% of the total UK population aged 0–14) was not included.

Overseen by a management committee that included epidemiologists, statisticians, paediatric oncologists/haematologists and expert scientists that was chaired by Professor Sir Richard Doll, 10 UKCCS administrative areas covering the whole of Britain were created for the purposes of data collection, with each being the responsibility of a well-established epidemiological research centre[8] (online supplemental figure 1).

The study began on 1 January 1991 in Scotland and on either 1 April 1992 or 1 September 1992 in the nine UKCCS administrative areas of England and Wales. In Scotland, case accrual ended on 31 December 1994, and in England and Wales, it was restricted to children diagnosed with leukaemia or non-Hodgkin's lymphoma throughout 1995 and leukaemia alone throughout 1996. In order to ensure ascertainment completeness, proactive cancer notification systems were established in all diagnostic/treatment centres across the UK, the majority of cases being notified at the point of diagnosis directly to the relevant UKCCS administrative centre by paediatric oncologists or haematologists practising in the study area. Subsequent crosschecks were made against regional childhood cancer registries (where they existed); the population-based National Registry of Childhood Tumours (NRCT), which covered England, Scotland and Wales[8 19]; and clinical trial datasets (acute lymphoblastic leukaemia).

With the permission of the child's clinical team, the parents of children with cancer were subsequently contacted. Each child whose parents agreed to be interviewed (3835/4430; 87%) was individually matched on sex, date of birth (month and year) and UKCCS region of residence to 10 controls who were randomly selected from the same population-based health service authority list as the case. With their general practitioner's (GP's) permission, the parents of the first two controls on the list ('first-choice' controls) were contacted and asked to participate in the study; but if the GP declined, or the parents did not wish to be interviewed, the next child was selected, and this procedure was repeated until two control families agreed to take part. By the end of the data collection phase, 96.7% (3786/3835) cases had two participating controls and 1.3% (49/3835) had one.

In order to monitor the characteristics of respondents and non-respondents across the study, and examine the potential impact of selection/participation bias, the registration details of all cases and controls were retained, regardless of whether or not their families were interviewed. Using standard methods, address postcodes were used to allocate deprivation scores for the home at diagnosis (all registered cases) or pseudo-diagnosis (all registered controls; date coinciding with the exact age that the corresponding case was diagnosed) and also at the time they were born (using mother's address on the child's birth certificate). Full details of the methods used have been previously published.[8 18] Briefly, deprivation categories were then derived by dividing the continuous score for the national 1991 census enumeration areas into five

**Table 1** Deprivation at cancer diagnosis/pseudo-diagnosis of subjects identified for inclusion on the case-control study and those for entry into the matched cohorts

| | Case-control study: subjects targeted for inclusion | | Five-year survivor cohorts | |
| --- | --- | --- | --- | --- |
| | Cases | First-choice controls* | Cases | First-choice controls |
| Deprivation | N (%) | N (%) | N (%) | N (%) |
| | 4430 (100) | 7658 (100) | 3125 (100) | 5606 (100) |
| Affluent 1 | 852 (19.2) | 1501 (19.6) | 613 (19.6) | 1094 (19.5) |
| 2 | 913 (20.6) | 1492 (19.5) | 666 (21.3) | 1111 (19.8) |
| 3 | 852 (19.2) | 1513 (19.8) | 620 (19.8) | 1080 (19.3) |
| 4 | 891 (20.1) | 1511 (19.7) | 604 (19.3) | 1138 (20.3) |
| Deprived 5 | 879 (19.8) | 1587 (20.7) | 604 (19.3) | 1140 (20.3) |
| Not known | 43 (1.0) | 54 (0.7) | 18 (0.6) | 43 (0.8) |
| | $\chi^2$=6.12, p=0.30 | | $\chi^2$=5.57, p=0.35 | |

*Controls were not selected for 595 cases (13.4%) whose parent(s) were not interviewed; and only one suitable first-choice control was identified for 12 of the remaining 3835 cases.

equally sized groups, with group 1 representing the most affluent and group 5 the most deprived.

Illustrating the UKCCS's potential to examine issues relating to selection bias, the first two columns of table 1 distribute all registered subjects in the case-control study according to the area-based deprivation score of their home at diagnosis (cases) or pseudo-diagnosis (controls; date coinciding with the exact age that the corresponding case was diagnosed). With roughly 20% of the population in each quintile, the deprivation distribution of the 5-year survivor cohorts (last two columns of table 1, right side) mirrors that of the case-control study (table 1, left side).

As expected, the deprivation distribution of those whose parents participated in the case-control study differed from those who did not (table 2). With 32.7% of non-participating controls residing in the most deprived quintile, the effect was more pronounced than in the case population (28.4%). Exacerbated by the

fact that participation among case families was higher (3835/4430=86.6%) than among first-choice control families (5526/7658=72.2%), the deprivation distributions of interviewed cases (n=3835, column 1) and first-choice controls (n=5526, column 3) differ significantly from each other ($\chi^2$=15.1, p=0.004). As expected, distributing individuals in the 5-year survivors cohort by the same categories produced similar differences (table 2, right side). Such categorisation will allow us not only to evaluate the potential impact of deprivation at birth and diagnosis on hospital activity among 5-year survivors but also to examine selection bias.

### Patient and Public Involvement and Engagement (PPIE)

The original case-control study was established over 30 years ago, at a time when lay-involvement in research was largely absent, and so did not benefit from formal links with patient/user groups. Members of the CCLG

**Table 2** Deprivation at cancer diagnosis/pseudo-diagnosis of subjects identified for inclusion on the case-control study and those for entry into the matched cohorts distributed according to whether the parents participated in the case-control study or not

| | Case-control study: subjects targeted for inclusion | | | | Five-year survivor cohorts | | | |
| --- | --- | --- | --- | --- | --- | --- | --- | --- |
| | Cases | | First-choice controls* | | Cases | | First-choice controls | |
| | Parents participated? | | Parents participated? | | Parents participated? | | Parents participated? | |
| | Yes | No | Yes | No | Yes | No | Yes | No |
| | N (%) | N (%) | N (%) | N (%) | N (%) | N (%) | N (%) | N (%) |
| Total | 3835(100) | 595(100) | 5526(100) | 2132(100) | 2819(100) | 306(100) | 4080(100) | 1526(100) |
| Deprivation | | | | | | | | |
| Affluent 1 | 774 (20.2) | 78 (13.1) | 1249 (22.6) | 252 (11.8) | 576 (20.4) | 37 (12.1) | 915 (22.4) | 179 (11.7) |
| 2 | 823 (21.5) | 90 (15.1) | 1181 (21.4) | 311 (14.6) | 618 (21.9) | 48 (15.7) | 873 (21.4) | 238 (15.6) |
| 3 | 773 (20.2) | 79 (13.3) | 1156 (20.9) | 357 (16.7) | 579 (20.5) | 41 (13.4) | 839 (20.6) | 241 (15.8) |
| 4 | 755 (19.7) | 136 (22.9) | 1051 (19.0) | 460 (21.6) | 531 (18.8) | 73 (23.9) | 798 (19.6) | 340 (22.3) |
| Deprived 5 | 710 (18.5) | 169 (28.4) | 889 (16.1) | 698 (32.7) | 515 (18.3) | 89 (29.1) | 655 (16.0) | 485 (31.8) |
| Not known | – | 43 (7.2) | – | 54 (2.5) | – | 18 (5.9) | – | 43 (2.8) |
| | $\chi^2$=64.8, p<0.001 | | $\chi^2$=355.2, p<0.001 | | $\chi^2$=43.3, p<0.001 | | $\chi^2$=238.5, p<0.001 | |

*Controls were not selected for 595 cases (13.4%) whose parent(s) were not interviewed; and only one first-choice control was identified for 12 of the remaining 3835 cases.

(Children's Cancer and Leukaemia Group, previously the UK Children's Cancer Study Group) were, however, fully involved throughout; and the study website (www.UKCCS.org) contains information about ongoing research activities and fair processing. The transformation of the study to focus on late effects among long-term childhood cancer survivors means that we can now move forwards to enable patients and the public to work in partnership with researchers and clinical staff to co-develop and deliver the research, disseminate findings and determine the direction of future embedded studies. This input will ensure data are used in a way that secures maximise impact, improves long-term care and promotes effective resource planning and management in the future.

## Follow-up

The fact that basic details (matching variables, postcode, NHS number) of all cases and controls were registered and retained, regardless of the families' participation, enabled the conversion of the study into a matched cohort that could track all registered cases (n=4430) and their corresponding first-choice controls (n=7658) who were targeted for interview (table 1; first two columns). In addition to the targeted case and control populations, replacement controls are being tracked, enabling the study to also make comparisons between the 3835 cases

(table 2, column 1) and 7621 controls (5526 first-choice (table 2, column 3) and 2095 replacement controls) whose parents participated.

Tracking all of the registered cases and controls, the UKCCS now operates on a legal basis that permits follow-up information to be obtained from NHS administrative healthcare records without explicit consent, enabling all cases and their corresponding controls to be 'flagged' at the national level and tracked via linkage to nationwide information on deaths, cancer registrations and hospital episode data (inpatient day and overnight admissions; outpatient appointments and visits; accident and emergency presentations and maternity admissions). Importantly, in line with current ethical and governance requirements, the matched cohorts are now pseudonymised; NHS identifiers are no longer held by the study but by the national administrative bodies supplying linked data (NHS England, Public Health Scotland, NHS Central Register Scotland, and the Secured Anonymised Information Linkage Databank).

Beginning with the numbers registered in the original case-control study, the flow diagram presented in figure 1 gives the number of cases and controls who survived 5 years or more, followed by the number of cases who were successfully linked to national administrative data

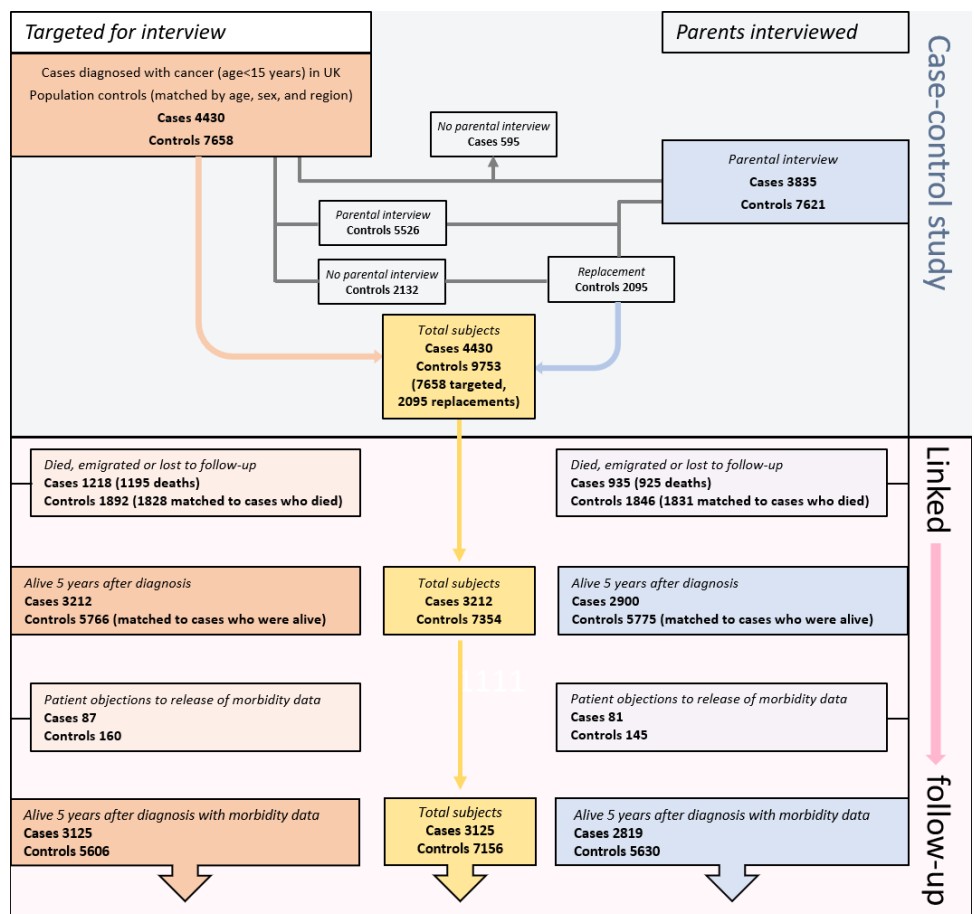

**Figure 1** UKCCS subjects distributed by parental interviewed status (total targeted and total interviewed) and follow-up status; the number of 5-year survivors with linked health data who are being followed is presented in the bottom row.

**Table 3** National administrative data available for comparative analysis: United Kingdom Childhood Cancer Study (UKCCS)

| Source of data | Description of fields |
| --- | --- |
| Case-control study | Sex; year of birth; date of diagnosis/pseudo-diagnosis; Townsend deprivation score of address at diagnosis/pseudo-diagnosis from FHSA/HB* registers |
| Birth certificates | Townsend deprivation score of address at birth |
| Cancer registrations | Date of diagnosis; date of registration; topography (ICD‡ revisions 7–10); morphology (ICD-O§ revisions 1–3) |
| Deaths and Emigrations | Date and causes of death (ICD-10‡); dates of emigrations and returns |
| Secondary Care Data (Hospital Activity – HES†) | |
| Inpatient and Day Cases | Date of admission; date of discharge; dates and types of procedures (OPCS-4¶, maximum 24); conditions at discharge (ICD-10‡, maximum 20); consultant specialties involved; source of referral; discharge destination; IMD** score for residence at admission |
| Outpatient | Date of appointment and attendance flag; dates of procedures (maximum 24); consultant specialties involved; IMD** score for residence at appointment |
| Accident and Emergency | Date and reason for attendance; investigations and treatments; source of referral; IMD** score for residence at attendance |
| Maternity Inpatient and Day Cases | Date of admission to obstetrics or midwifery; date of discharge; conditions at discharge (ICD-10‡ maximum 20); age at birth; delivery method; babies' birth weights; IMD** score for residence at admission |

*FHSA/HB: Family Health Strategic Authority or Health Board.
†HES: Hospital Episode Statistics in England, or equivalently, Scottish Morbidity Record (SMR) in Scotland, and the Patient Episode Database for Wales (PEDW).
‡ICD: International Classification of Diseases.
§ICD-O: International Classification of Diseases for Oncology.
¶OPCS-4: Office of Population Censuses and Surveys Classification of Interventions and Procedures version 4.
**IMD: Index of Multiple Deprivation

sources. Cases enter the survivors' cohort exactly 5 years after their cancer was first diagnosed and controls 5 years after their corresponding case's cancer was first diagnosed (pseudo-diagnosis date). Controls who emigrated, died or were lost to follow-up before 5 years had elapsed are not included in the survivor cohort, nor are controls who were matched to cases who died within 5 years of their diagnosis, since, where applicable, their follow-up was truncated when their matched case died (figure 1; bottom row). All those who were targeted for interview in the original case-control study are shown on the left side of figure 1 and all those whose parents were interviewed on the right side; the total number of individuals that are being followed is shown in the yellow boxes in the centre of the figure.

With respect to the available data, the time periods currently covered by the various linked healthcare datasets, which are updated on an annual basis, are shown in online supplemental figure 2; and the key data domains and fields are listed in table 3.

### Characteristics of individuals in the study

Detailed diagnostic and treatment information was obtained from multiple sources, including diagnostic/treatment centres, national treatment trials (where they existed), hospital records and reference laboratories. The diagnostic distribution, coded to the International Classification of Childhood Cancer Third Edition (ICCC-3), as well as information on age, sex and interview status, is shown in table 4. As expected, given the increasing emphasis on leukaemia in the later years of the case-control study's data collection phase, the final diagnostic distribution was weighted towards leukaemia. Variations with parental interview status are also evident, with the non-interviewed group containing proportionally more CNS tumours (ICCC III) and other cancers (ICCC XI and XII). Among all registered cases, 1195 (27.0%) died within the first 5 years, and while in total, over 70% of the cases were alive at 5 years after diagnosis, the proportions within each ICCC-3 group varied with cancer (table 5).

### FINDINGS TO DATE

The UKCCS provides a valuable resource within which to examine both the potential causes and long-term health consequences of childhood cancer. Thus far, over 90 peer-reviewed articles containing UKCCS data have been published, and a full list is available on the study website (www.UKCCS.org). Assembling data from multiple sources, key reports include ALL genome-wide association studies (GWAS),[14 20] and those examining the potential aetiological roles of a wide range of biological (eg, infections/markers of infectious exposure),[21–23] physical (eg, ionising/non-ionising radiation)[12 13 24 25] and chemical (eg, prescription drugs/smoking) agents acting on parents before the child's conception and the child during pregnancy/early life,[10 26 27] as well as relationships with other illnesses (eg, allergy) and birth characteristics (eg, birth weight).[21 28 29] Importantly, the UKCCS accessed medical records to help answer key questions relating to prescription drugs, illness histories and comorbidities; such contemporaneously recorded information, which by its nature is free from differential recall bias, often producing very different results from that obtained by self-report.[11 22 30 31]

With respect to the longer term health and healthcare needs of childhood cancer survivors, the conversion of the UKCCS into a matched cohort is now yielding

**Table 4** Diagnostic frequencies, median ages at diagnosis and parental interview status of children (0–14 years) registered in the United Kingdom Childhood Cancer Study (UKCCS)

| Diagnosis (ICCC group) | Total N (%) | Males (%) | Age, median (IQR) | Interviewed | |
|---|---|---|---|---|---|
| | | | | Yes N (%) | No N (%) |
| All cancers | 4430 (100.0) | 55.8 | 5.1 (2.7–9.7) | 3835 (100.0) | 595 (100.0) |
| Leukaemia (I) | 1911 (43.1) | 55.8 | 4.5 (2.8–8.0) | 1734 (45.2) | 177 (29.7) |
| Acute lymphoblastic leukaemia (Ia) | 1580 (35.7) | 55.9 | 4.4 (2.9–7.6) | 1462 (38.1) | 118 (19.8) |
| Acute myeloid leukaemia (Ib) | 294 (6.6) | 53.7 | 5.3 (1.8–11.2) | 248 (6.5) | 46 (7.7) |
| Lymphomas & reticuloendothelial (II) | 413 (9.3) | 72.9 | 9.9 (5.7–12.9) | 357 (9.3) | 56 (9.4) |
| Hodgkin lymphoma (IIa) | 132 (3.0) | 71.2 | 11.6 (7.8–13.5) | 117 (3.0) | 15 (2.5) |
| Non-Hodgkin's lymphoma (IIb) | 258 (5.8) | 72.9 | 9.3 (5.2–12.0) | 226 (5.9) | 32 (5.4) |
| CNS, intracranial and intraspinal (III) | 848 (19.1) | 50.1 | 6.5 (3.3–10.1) | 686 (17.9) | 162 (27.2) |
| Ependymoma and choroid plexus (IIIa) | 101 (2.3) | 58.4 | 3.4 (1.4–9.0) | 90 (2.3) | 11 (1.8) |
| Astrocytoma (IIIb) | 372 (8.4) | 44.1 | 6.9 (4.0–10.5) | 303 (7.9) | 69 (11.6) |
| Intracranial and intraspinal embryonal (IIIc) | 189 (4.3) | 55.6 | 6.0 (2.7–8.8) | 162 (4.2) | 27 (4.5) |
| Neuroblastoma and other peripheral neural (IV) | 222 (5.0) | 59.5 | 1.8 (0.7–3.7) | 188 (4.9) | 34 (5.7) |
| Retinoblastoma (V) | 112 (2.5) | 55.4 | 1.4 (0.7–2.7) | 87 (2.3) | 25 (4.2) |
| Renal (VI) | 221 (5.0) | 51.6 | 3.2 (1.8–5.2) | 199 (5.2) | 22 (3.7) |
| Hepatic (VII) | 36 (0.8) | 58.3 | 1.5 (0.9–3.3) | 32 (0.8) | 4 (0.7) |
| Bone (VIII) | 121 (2.7) | 53.7 | 11.1 (8.9–13.4) | 106 (2.8) | 15 (2.5) |
| Soft tissue and other sarcomas (IX) | 266 (6.0) | 55.3 | 5.7 (2.8–10.6) | 233 (6.1) | 33 (5.5) |
| Germ cell and gonadal (X) | 114 (2.6) | 54.4 | 6.9 (1.4–12.2) | 98 (2.6) | 16 (2.7) |
| Other cancers (XI and XII) | 99 (2.2) | 43.4 | 10.6 (6.7–13.6) | 62 (1.6) | 37 (6.2) |

important results about the morbidity and mortality of childhood cancer survivors. Early findings based on the case population alone highlighted survival variations with age and sex across a range of subtypes,[15] and for acute lymphoblastic leukaemia, survival associations with area-based measures of deprivation at the time of diagnosis were also noted.[16] With additional years of follow-up, we are now able to report a 25-year survival of 65.3% for all cancers combined (95% CI 63.8–66.7). More recently, we have begun examining secondary care hospital activity patterns, comparing findings in the case population to those in the control population over the first 25 years. Adding to excess risks of death and cancer, our first report demonstrated that survivors of childhood acute lymphoblastic leukaemia (ALL) experienced excess outpatient and inpatient activity across their teenage and young adult years, which was not related to routine follow-up monitoring. Indeed, with hospital activity being higher than expected in specialties covering most organ and tissue systems, survivors were more than twice as likely to fall under the care of endocrinology, cardiology, respiratory medicine, ophthalmology, neurology and/or gastroenterology specialists.[17] Furthermore, these differences showed no signs of diminishing in the first 25 years of follow-up, underscoring the need to take prior cancer drug and/or radiation treatment into account when interpreting seemingly unrelated symptoms in later life.

## STRENGTHS AND LIMITATIONS

Covering England, Scotland and Wales, findings from this population-based age- and sex-matched cohort study are based on routinely compiled linked health data, not self-report. With minimal selection bias and loss to follow-up, the matched-cohort design allows more granular analyses to be performed than is possible in case cohorts that rely solely on general population rates, as has been done in other UK-based studies.[32–35] For example, at the point of cohort entry, the individually age- and sex-matched control population has a similar deprivation distribution to the case population, providing a robust baseline against which to evaluate the potential impact of deprivation across the life-course. In addition, because the UKCCS cohorts can be divided into those whose families participated in the original case-control study and those who did not, the study is well-placed to investigate the impact of selection bias: an analysis that most cohorts cannot undertake, largely because they were not designed to do so, most being either record-based (as on the left side of figure 1)[36–39] or interview-based (as on the right side of figure 1).[40 41] Thus far, with follow-up extending over 25 years from diagnosis, we have examined health through the teenage and early adult years, and individuals in the matched cohorts will continue to be tracked as they age. With respect to limitations, lack of reliable details about ethnicity as well as information from primary care and psychosocial services are obvious deficiencies that currently affect all UK record-linkage

**Table 5** Number of children in the case-control study, numbers alive 5 years after diagnosis and current duration of follow-up (July 2020 England and Wales; June 2022 Scotland) by diagnosis (ICCC-3): UKCCS

| Diagnosis (ICCC group) | Total | Alive 5 years after diagnosis* | | Total person-years† | |
|---|---|---|---|---|---|
| | | N | % | Median (IQR) | Total |
| Total cases | 4430 | 3212 | 72.5 | 25.3 (3.5–27.1) | 78 623 |
| Leukaemias (I) | 1911 | 1448 | 75.8 | 24.7 (5.2–26.7) | 34 423 |
| Acute lymphoblastic leukaemia (Ia) | 1580 | 1275 | 80.7 | 24.9 (9.0–26.9) | 30 206 |
| Acute myeloid leukaemia (Ib) | 294 | 161 | 54.8 | 19.2 (0.9–25.8) | 3913 |
| Lymphomas and reticuloendothelial (II) | 413 | 344 | 83.3 | 26.1 (20.4–27.5) | 8434 |
| Hodgkin lymphoma (IIa) | 132 | 127 | 96.2 | 27.3 (26.1–28.2) | 3194 |
| Non-Hodgkin's lymphoma (IIb) | 258 | 203 | 78.7 | 25.4 (12.9–27.1) | 4880 |
| CNS, intracranial and intraspinal (III) | 848 | 534 | 63.0 | 23.3 (1.3–26.9) | 13 021 |
| Ependymoma and choroid plexus (IIIa) | 101 | 62 | 61.4 | 9.2 (1.9–26.5) | 1359 |
| Astrocytoma (IIIb) | 372 | 275 | 73.9 | 26.2 (3.5–27.4) | 6894 |
| Intracranial embryonal (IIIc) | 189 | 84 | 44.4 | 3.0 (0.8–26.1) | 2050 |
| Neuroblastoma and other peripheral neural (IV) | 222 | 121 | 54.5 | 11.6 (1.3–26.8) | 3109 |
| Retinoblastoma (V) | 112 | 108 | 96.4 | 26.8 (26.0–27.5) | 2769 |
| Renal (VI) | 221 | 173 | 78.3 | 26.1 (19.3–27.4) | 4477 |
| Hepatic (VII) | 36 | 27 | 75.0 | 25.8 (2.6–26.9) | 674 |
| Bone (VIII) | 121 | 70 | 57.9 | 20.0 (1.9–26.9) | 1799 |
| Soft tissue and other sarcomas (IX) | 266 | 167 | 62.8 | 24.7 (1.8–27.4) | 4205 |
| Germ cell and gonadal (X) | 114 | 92 | 80.7 | 26.5 (17.5–27.5) | 2359 |
| Other cancers (XI and XII) | 99 | 70 | 70.7 | 26.5 (2.4–27.7) | 1793 |
| Total controls | 9753 | 9685 | 99.3 | 26.5 (25.2–27.6) | 243 928 |
| All first choice | 7658 | 7594 | 99.2 | 26.6 (25.2–27.6) | 191 351 |
| Replacements (parents interviewed) | 2095 | 2091 | 99.8 | 26.5 (25.1–27.5) | 52 577 |

*97.7% of cases and 97.1% of controls have been traced and have administrative data.
†Person-years are calculated for the total number of cases from diagnosis to end of follow-up, and for controls from their pseudo-diagnosis date, that is the date they were the same age as their matched case was when they were diagnosed.

studies of the type described here.[42–44] Analyses are also constrained by the fact that national administrative healthcare data are primarily collected for administrative and clinical purposes and not for research.

## COLLABORATION
To protect privacy and confidentiality, approval for the linkage of the UKCCS to health data is provided under strict conditions for the storage, retention and use of the data. The current approvals permit storage of the data at one site (University of York) for use by individually named researchers. We encourage interested parties to contact us to discuss potential analyses. Reasonable requests for data access can be submitted to the study Principal Investigator, ER (eve.roman@york.ac.uk), for review by the UKCCS investigator team. The PI will contact the relevant agencies to explore the feasibility of data sharing subject to ethical and data access agreements; should data sharing be agreeable with the agencies, approval for data access and specific analyses will be required from the appropriate ethics committees and data sharing agreements put in place.

**Author contributions** ER contributed to the design of the original case-control study, initiated the matched cohort, drafted the report and is the guarantor.

Contributing to both the original case-control and the matched cohort, SK and JS have also been involved with the UKCCS since its inception. EK now leads on UKCCS data management and was involved in the design of the matched cohort. AB, EK, JS and AS have conducted many UKCCS analyses, and SK is the clinical lead for the study. DH and RS lead on PPIE and have extensive experience of working with patients. All authors contributed to the written report.

**Funding** The work reported on here was previously supported by Blood Cancer UK grant number 15037, and is jointly now supported by Blood Cancer UK and Cancer Research UK (grant number 29685). AB is supported by a fellowship from the Fondation ARC pour la Recherche sur le Cancer (PDF20190508759). ER and AS are supported in part by the National Institute for Health and Care Research (NIHR) Leeds Biomedical Research Centre. The funders had no role in considering the study design or in the collection, analysis, interpretation of data, writing of the manuscript or the decision to submit the article for publication.

**Map disclaimer** The inclusion of any map (including the depiction of any boundaries therein), or of any geographic or locational reference, does not imply the expression of any opinion whatsoever on the part of BMJ concerning the legal status of any country, territory, jurisdiction or area or of its authorities. Any such expression remains solely that of the relevant source and is not endorsed by BMJ. Maps are provided without any warranty of any kind, either express or implied.

**Competing interests** None of the authors have any conflicts of interest

**Patient and public involvement** Patients and/or the public were not involved in the design, or conduct, or reporting, or dissemination plans of this research.

**Patient consent for publication** Not applicable.

**Ethics approval** The United Kingdom Childhood Cancer Study (UKCCS) has ethics approval from the Yorkshire & the Humber-Leeds West Research Ethics Committee (reference 18/YH/0135) and exemption from Section 251 of the Health & Social Care Act (reference 18/CAG/0066).

**Provenance and peer review** Not commissioned; externally peer reviewed.

**DATA SHARING** Reasonable requests for data sharing will be considered by the authors.

**ORCID iDs**
Eve Roman http://orcid.org/0000-0001-7603-3704
Eleanor Kane http://orcid.org/0000-0002-7438-9982
Alexandra Smith http://orcid.org/0000-0002-1111-966X
Debra Howell http://orcid.org/0000-0002-7521-7402
Rebecca Sheridan http://orcid.org/0000-0002-7715-1224

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
