## [Reviewer comments · BMJ Open]

ARTICLE DETAILS

TITLE (PROVISIONAL)	Cohort Profile: The United Kingdom Childhood Cancer Study (UKCCS): a UK-wide population-based study examining the health of cancer survivors
AUTHORS	Roman, Eve; Kane, Eleanor; Smith, Alexandra; Howell, Debra; Sheridan, Rebecca; Simpson, Jill; Bonaventure, Audrey; Kinsey, Sally

VERSION 1 – REVIEW

REVIEWER	Peh J. Ho National University Singapore Saw Swee Hock School of Public Health
REVIEW RETURNED	25-Apr-2023

GENERAL COMMENTS	Thank you for the opportunity to review this manuscript. The authors have shown the potential of setting-up a population-based study with long-term follow-up. The data is valuable to understand the impact of childhood cancer and deprivation on development. Minor comments In the section COHORT DESCRIPTION 1) paragraph 1. It is unclear from the text if the entire population (England, Scotland, and Wales) divided into ten areas or were ten areas selected? 2) paragraph 2. Spelling out the month will increase readability. 3) paragraph 3. Controls were age- and sex-matched. However, was ethnicity considered? Would the percentage of the majority ethnicity be presented. It would be informative to discuss the potential effect of ethnicity on participation and outcomes. In the section FINDINGS TO DATE 4) It would be informative to include the 20- or 25-year survival in this manuscript and cite the source if it is published.
--

REVIEWER	Christel Häggström Uppsala University, Department of Surgical Sciences
REVIEW RETURNED	04-May-2023

GENERAL COMMENTS	General comments Very well written and clear cohort profile. I have no major comments, only a few very minor comments. Minor comments As this is a national study (and "UK-wide"): Is there any cases of childhood cancer that are not included in the study? Capture
---

	rate/coverage of the original UKCCS would be nice to know, if that data is available on a national level. Furthermore, as there are no data from Northern Ireland, can you really call it national? This should at least be mentioned as a limitation. Maybe one sentence can be added that more clearly state the aim in this extension of the original project and possibly also some examples mortality and morbidity that will be studied, in the introduction section. The first-choice and replacement control selection were well described. One question that comes up is if any of the participated cases that had no control selected (if all 10 of the selected controls denied participation). Was the case excluded then? Also, was there any cases with only one control? Abbreviations in the abstract not written out at first use, please correct this. Table 2; does the age refer to age at diagnosis, or age at interview, or some other age? A very minor suggestion: Maybe the authors can think about the avoiding the word “case-cohort” to avoid confusion with that study design. I suggest to use case population and control/reference population or something similar. Please doublecheck that the wording of these populations are consistent across the full text incl figures/tables. I also miss the heading for collaboration (included in the submission requirements).
--	---

REVIEWER	Biljana Gigic Heidelberg University, Department of General, Visceral and Transplantation Surgery
REVIEW RETURNED	17-May-2023

GENERAL COMMENTS	This paper describes The United Kingdom Childhood Cancer Study (UKCCS), a UK-wide population-based study examining the longer-term morbidity and mortality of cancer survivors diagnosed before 15 years of age. There is a substantial amount of detail with regard to cohort description, but the paper appears somewhat sparse with regard to results. Furthermore, the study design has been previously published at the British Journal of Cancer in 2000, the novelty of the current matched cohort needs to be described more in detail. Specific comments to be addressed: In general, some of the sentences are very long and convoluted which compromises the readability of the manuscript, i.e. abstract, line 22-28. ABSTRACT The abbreviations within the abstract (e.g. line 4, line 25, line 26) should be spelled out. 1) Purpose: The abstract would benefit by including a more detailed description of the aims of the new study design. Please revise accordingly. 2) Future plans: The authors have proposed an item “Future plans” in the Abstract. This only provides a brief description, and I don’t think I have seen a dedicated Section in the manuscript. The authors did give a few leads here and there, but they could perhaps provide further details (e.g., planned investigations, statistical analyses etc.). INTRODUCTION
---

	1) The authors need to further elaborate on the novelty of this particular study since the study design has been previously published. 2) Within the introduction, there is a focus on treatment, treatment response as well as treatment-related adverse health problems. Within the data collection in Box 1 it is not visible that treatment and treatment-related events have been abstracted. Are treatment-related health problems an aim of this study? The author should be more precise what the focus is and what has been already done in the field of this research area. 3) Please clearly state aims and objectives. 4) In general, the introduction is not comprehensive enough. The authors should also elaborate on causes of childhood cancer, childhood cancer survival, and health of childhood cancer survivors, as seen on the UKCCS website. 5) There is a typo in line 9. COHORT DESCRIPTION 1) The period of the study presented in the report differs from the period that has been reported in the British Journal of Cancer in 2000. Please clarify. 2) In regard to selection bias, it is important to illustrate differences between parents who did not participate and those who did. The authors should address this. 3) The initial study design was a case control study, in this cohort profile, it seems like the study has been converted to a matched cohort. It is hard to understand the novelty and the differences between the old and the new study design. The authors should present this more clearly. DATA SHARING 1) The authors stated that reasonable requests for data sharing will be considered. Would the authors further elaborate on how collaborators could have access to data from this study?
--	---

VERSION 1 – AUTHOR RESPONSE

Reviewer: 1

Dr. Peh J. Ho, National University Singapore Saw Swee Hock School of Public Health

Comments to the Author:

Thank you for the opportunity to review this manuscript. The authors have shown the potential of setting-up a population-based study with long-term follow-up. The data is valuable to understand the impact of childhood cancer and deprivation on development.

Minor comments

In the section COHORT DESCRIPTION

1) paragraph 1. It is unclear from the text if the entire population (England, Scotland, and Wales) divided into ten areas or were ten areas selected?

Response: The text has been changed to clarify that the 10 regions covered the whole of Britain and a new Supplementary Figure added (Supplementary Figure 1).

2) paragraph 2. Spelling out the month will increase readability.

Response: All dates have been changed to this format.

3) paragraph 3. Controls were age- and sex-matched. However, was ethnicity considered? Would the percentage of the majority ethnicity be presented. It would be informative to discuss the potential effect of ethnicity on participation and outcomes.

Response: We agree that ethnicity would be an interesting variable to consider; unfortunately, however, it is not routinely recorded in the population registers that were used for subject selection, and it is not recorded at birth registration. Further although information about parental ethnic group was requested from participants at interview (obviously not available for non-participants), it was unreliably recorded and the variable was not used in any analyses. For information, the 1991 UK census was the first one to include a question on ethnic group, revealing that at that time around 95% identified as “white”.

In the section FINDINGS TO DATE

4) It would be informative to include the 20- or 25-year survival in this manuscript and cite the source if it is published.

Response: Having previously reported up to 15-year survival (reference 9), we are now able to estimate 25-year survival; as such we have added the following to the penultimate paragraph: *“With additional years of follow-up, we are now able to report a 25-year survival of 65.3% for all cancers combined (95% confidence interval 63.8-66.7).”*

Reviewer: 2

Dr. Christel Häggström, Uppsala University, Umeå University

Comments to the Author:

General comments

Very well written and clear cohort profile. I have no major comments, only a few very minor comments.

Minor comments

As this is a national study (and “UK-wide”): Is there any cases of childhood cancer that are not included in the study? Capture rate/coverage of the original UKCCS would be nice to know, if that data is available on a national level. Furthermore, as there are no data from Northern Ireland, can you really call it national? This should at least be mentioned as a limitation.

Response: To ensure completeness, at the end of the case-control data collection phase, cross-checks were made against regional childhood cancer registries, and the National Registry of Childhood Tumours. The proactive case ascertainment methods in the UKCCS were found to be highly complete (UKCCS investigators 2000, Smith et al 2006). The following sentence and a further reference have been added to the Cohort Description at the end of the second paragraph. *“ Subsequent crosschecks were made against regional childhood cancer registries (where they existed), the population-based National Registry of Childhood Tumours (NRCT), which covered England, Scotland and Wales [8,14], and clinical trial datasets (acute lymphoblastic leukaemia only).”*

With respect to cancer exclusions, Scotland did not include children diagnosed with retinoblastoma.

With regard to population exclusions, Northern Ireland (~3.5% of the total UK 0-14 population in 1991), was omitted for pragmatic/logistical reasons. This decision was taken following discussion with all relevant stake-holders, but UK label was retained, again with the support of all stakeholders. The distinction between the UK and Britain is now flagged more overtly in paragraph 1 of the Cohort Description and the following text added: *“...; unfortunately for logistical reasons Northern Ireland (~3.5% of the total UK 0-14 population) was not included.”*

Maybe one sentence can be added that more clearly state the aim in this extension of the original project and possibly also some examples mortality and morbidity that will be studied, in the introduction section.

Response: Thank you for your suggestion. The second paragraph of the Introduction has now been modified in several places, please see document.

The first-choice and replacement control selection were well described. One question that comes up is if any of the participated cases that had no control selected (if all 10 of the selected controls denied participation). Was the case excluded then? Also, was there any cases with only one control?

Response: Good question – all had a least one; 3786/3835 (98.7%) had two and 49 (1.3%) had 1. We have added the following to the Cohort Description; “*By the end of the data collection phase, 96.7% (3786/3835) cases had two participating controls and 1.3% (49/3835) had one.*”

Abbreviations in the abstract not written out at first use, please correct this.

Response: This has been corrected

Table 2; does the age refer to age at diagnosis, or age at interview, or some other age?

Response: Our apologies, this was an omission- we have now modified the Table’s title to show that it is age at diagnosis (now Table 3).

A very minor suggestion: Maybe the authors can think about the avoiding the word “case-cohort” to avoid confusion with that study design. I suggest to use case population and control/reference population or something similar. Please doublecheck that the wording of these populations are consistent across the full text incl figures/tables.

Response: Thank you for the helpful suggestion; we can confirm that this has now been applied.

I also miss the heading for collaboration (included in the submission requirements).

Response: Our apologies for omitting this requirement. This section has now been added (see response to the Editor).

Reviewer: 3

Dr. Biljana Gigic, Heidelberg University

Comments to the Author:

This paper describes The United Kingdom Childhood Cancer Study (UKCCS), a UK-wide population-based study examining the longer-term morbidity and mortality of cancer survivors diagnosed before 15 years of age. There is a substantial amount of detail with regard to cohort description, but the paper appears somewhat sparse with regard to results. Furthermore, the study design has been previously published at the British Journal of Cancer in 2000, the novelty of the current matched cohort needs to be described more in detail.

Specific comments to be addressed:

In general, some of the sentences are very long and convoluted which compromises the readability of the manuscript, i.e. abstract, line 22-28.

Response: Thank you for pointing this out. The Abstract has been modified to improve its readability, and this part now reads: “*More recently, comparing the health-activity patterns of the case-cohort and control-cohort we found that survivors of childhood ALL experienced excess outpatient and inpatient activity across their teenage/young adult years. Adding to increased risks of cancer and death and involving most clinical specialties, excesses were not related to routine follow-up monitoring, and showed no signs of diminishing over time*”

ABSTRACT

The abbreviations within the abstract (e.g. line 4, line 25, line 26) should be spelled out.

Response: *Abbreviations in the abstract have now been spelt out.*

1) Purpose: The abstract would benefit by including a more detailed description of the aims of the new study design. Please revise accordingly.

Response: Addressing this and other queries, the 300-word Abstract has been modified in several places, including Future Plans (see below).

2) Future plans: The authors have proposed an item “Future plans” in the Abstract. This only provides a brief description, and I don’t think I have seen a dedicated Section in the manuscript. The authors did give a few leads here and there, but they could perhaps provide further details (e.g., planned investigations, statistical analyses etc.).

Response: Future plans is a journal requirement for the Abstract i.e. not one we have proposed. This section now reads “*With annual linkage updates, the UKCCS’s maturing population-based matched-cohorts provide the foundation for tracking the health of individuals through their lifetime. Comparing the experience of childhood cancer survivors to that of unaffected general-population counterparts, this will include examining subsequent morbidity and mortality, secondary care hospital activity, and the impact of deprivation on longer-term outcomes.*”

INTRODUCTION

1) The authors need to further elaborate on the novelty of this particular study since the study design has been previously published.

Response: This report expands the previously published description of the case-control study to describe how the study is tracking subjects in order to study the health of childhood cancer survivors. We apologise if this was not sufficiently clear, Figure 1 has now been updated and the text has been modified in several places throughout the report.

2) Within the introduction, there is a focus on treatment, treatment response as well as treatment-related adverse health problems. Within the data collection in Box 1 it is not visible that treatment and treatment-related events have been abstracted. Are treatment-related health problems an aim of this study? The author should be more precise what the focus is and what has been already done in the field of this research area.

Response: We apologise for any confusion caused; treatment-related data were not collected as part of the original case-control study (conducted in the 1990s) and no population-based electronic registers exist. Indeed, the only treatment data we have access to relates to individuals with acute lymphoblastic leukaemia (ALL) who were recruited into clinical trials (predominantly UKALL XI which ran from 1992-6); these data have been well described by others and were primarily used here for cross-checking purposes (see response to Reviewer 2). Obviously, although all survivors are at risk of late treatment effects, treatment-related health problems at the individual level are not a major focus of this cohort, and anything we observe in the future will need to be examined carefully in studies that have access to individual treatment data.

3) Please clearly state aims and objectives.

Response: The Introduction has been modified, and the purpose of the study more clearly stated.

4) In general, the introduction is not comprehensive enough. The authors should also elaborate on causes of childhood cancer, childhood cancer survival, and health of childhood cancer survivors, as seen on the UKCCS website.

Response: This cohort profile outlines the methods and baseline data underpinning the matched cohorts, and detailed discussion about results are not included. The causes of childhood cancer were the focus of the case-control study; they are not the focus of the cohort and so are not included in this profile (this accords with journal guidelines). We have, however, restructured the Introduction and included a several more general statements along the lines requested. The following text has been added: “...*Collecting data from multiple sources (including interviews with parents, primary care records, obstetric/neonatal notes, birth certificates, household radiation measurements, and pre-treatment and remission blood samples), the case-control study investigated the potentially carcinogenic effects of a wide-range of physical (e.g. non-ionizing radiation), chemical (e.g. drugs) and biological (e.g. infectious agents) agents [8–14]. Examining associations in the pre-natal, in-utero, and post-natal periods, as well as associations with birth characteristics (e.g. birthweight) and other illnesses (e.g. allergies), over 90 reports have thus far been published (www.UKCCS.org).*”

5) There is a typo in line 9.

Response: Thank you for spotting that we had written “therapies improves” rather than “therapies improve”; we have now corrected our error.

COHORT DESCRIPTION

1) The period of the study presented in the report differs from the period that has been reported in the British Journal of Cancer in 2000. Please clarify.

Response: The period of case ascertainment in the study is from 1991 to 1996 as described in this paper and the materials- cases section of the BJC 2000 paper; the BJC abstract states to 1998 as

this is when data collection in the original case-control study ended rather than the diagnosis date of the last cases. We hope this clarifies the differences and we can only apologize for the confusion.

2) In regard to selection bias, it is important to illustrate differences between parents who did not participate and those who did. The authors should address this.

Response: Thank you for this observation. Table 1 has now been split into two (Tables 1 and 2), and additional text has been added. The text has also been modified, and we hope that this has clarified the situation.

3) The initial study design was a case control study, in this cohort profile, it seems like the study has been converted to a matched cohort. It is hard to understand the novelty and the differences between the old and the new study design. The authors should present this more clearly.

Response: We thank the author for their useful comment. We hope the changes we have made as a result of all the reviewers' comments have clarified the novelty, and explain how the study (which was originally designed to examine the potential causes of childhood cancer), can be utilised to investigate childhood cancer survivorship. We have also modified the flow diagram (Figure 1), which more clearly separates the earlier case-control study from the derivation of the matched cohorts.

DATA SHARING

1) The authors stated that reasonable requests for data sharing will be considered. Would the authors further elaborate on how collaborators could have access to data from this study?

Response: Our apologies for omitting this requirement. We have now added the COLLABORATION section; please see our response to the equivalent comment from the Editor and Reviewer 2.